# 2-Hydroxy-4-Methoxybenzaldehyde (2H4MB): Integrating Cell Culture, Metabolic Engineering, and Intelligent Genome Editing

**DOI:** 10.3390/ijms27010503

**Published:** 2026-01-03

**Authors:** Fatima Firdaus, Vikas Yadav, Muthusamy Ramakrishnan, Adla Wasi, Irfan Bashir Ganie, Anamica Upadhyay, Anwar Shahzad, Zishan Ahmad

**Affiliations:** 1Department of Chemistry, University of Lucknow, Lucknow 226007, India; fatima11012025@gmail.com; 2Department of Botany, Aligarh Muslim University, Aligarh 202002, India; yadavvikas535@gmail.com (V.Y.); adlawasi97@gmail.com (A.W.); irfanbashir301@gmail.com (I.B.G.); 3State Key Laboratory of Tree Genetics and Breeding, Co-Innovation Centre for Sustainable Forestry in Southern China, Bamboo Research Institute, Key Laboratory of National Forestry and Grassland Administration on Subtropical Forest Biodiversity Conservation, School of Life Sciences, Nanjing Forestry University, Nanjing 210037, China; ramky@njfu.edu.cn; 4Botany Department, School of Life Sciences, Dr. Bhimrao Ambedkar University, Agra 282007, India; anamicaupadhyay@gmail.com

**Keywords:** 2-Hydroxy-4-Methoxybenzaldehyde (2H4MB), in vitro culture, synthetic biology, metabolic engineering, secondary metabolites

## Abstract

2-Hydroxy-4-Methoxybenzaldehyde (2H4MB) is a valuable aromatic compound with applications in flavour, fragrance, and pharmaceuticals. Because of its endangered status and root-specific accumulation, its production in native plants is restricted. In order to increase 2H4MB yield, this study emphasises recent developments in metabolic engineering, synthetic biology, in vitro culture methods, and AI-assisted route prediction. This review discussed about how CRISPR-based genome editing can be used to modify important biosynthetic genes and regulatory components, as well as how predictive machine learning techniques can be used to improve production conditions. Inadequate genetic resources, poorly understood biosynthetic pathways, and a dearth of reliable transformation systems are among the present constraints. The work highlights the importance of using integrative plant biotechnology techniques to fully realise the industrial and medicinal potential of this underutilised chemical.

## 1. Introduction

Secondary metabolites from plants are still a useful source of bioactive substances with uses in industry, agriculture, and medicine [1]. The global herbal extract market size was estimated at USD 43.38 billion in 2024 [https://www.precedenceresearch.com/herbal-extract-market, accessed on 29 October 2025]. Nowadays, around 80% of the world’s population uses plants and their extracts for primary healthcare. The World Health Organisation projects that by 2050, the global herbal market would be worth USD $5 trillion [2]. Future large-scale cultivation systems have a lot of potential due to the rising demand for plant-based medications worldwide and the growth of trade [3,4,5]. Among these, 2-Hydroxy-4-Methoxybenzaldehyde (2H4MB) has drawn interest increasingly due to its extensive pharmacological potential, which includes anti-inflammatory, anticancer, antibacterial and antioxidant qualities [6,7] (Table 1). In terms of structure, 2H4MB is an isomer of vanillin (4-hydroxy-3-methoxybenzaldehyde), one of the most prized flavour molecules in the world’s food and fragrance industries [8]. Although vanillin is a standard molecule for the biotechnological synthesis of aromatic aldehydes, its structural resemblance to 2H4MB offers a solid biochemical basis for investigating similar biosynthetic and metabolic engineering strategies [9].

Many plants have reported 2H4MB, including member of the genus *Decalepis* (*D. hamiltoni*, *D. arayalpathra*, *D.salicifolia*) and *Hemidesmus indicus*, which is considered a rich source [10,11]. In most cases, 2H4MB mostly accumulates in the roots, requiring plant harvesting in order to extract it. Natural populations are seriously threatened by this technique, underscoring the necessity for alternate, sustainable production methods. The production of 2H4MB is believed to originate from the phenylpropanoid route, which entails a series of hydroxylation, O-methylation, and oxidation events mediated by significant enzymes such O-methyltransferases, hydroxylases, and dehydrogenases [12]. However, efforts to enhance natural production are limited because the exact enzymatic mechanism and genetic regulation are still mostly unknown.

**Table 1 ijms-27-00503-t001:** Biological activities of 2-Hydroxy-4-Methoxybenzaldehyde (2H4MB).

Plant	Plant Parts	Biological Role	Extracting Solvent	Test Organism Models/Cell/Tissue	References
*H. indicus*	Root	Anti-biofilm	Methanol	Human cell lines	[13]
*Mondia whytei*	Root	Ovicidal	NS	Anopheles gambiae Eggs	[14]
*H. indicus*	Root	TLR2 inhibition	Methanol	Synovial fibroblast and endothelial	[15]
*H. indicus*	Root	Anti-virulence	Methanol	Human peripheral blood mononuclear cells (PBMCs)	[16]
*H. indicus*	Root	Anti-carcinogenic	Methanol	Breast cancer	[17]
*D. arayalpathra*	Root	Antioxidant and anticancer	Dichloromethane	Breast cancer cells	[18]
*D. hamiltonii*	Tuber	Antioxidant	Aqueous	*C. elegans*	[19]
*Janakia arayalpatra*	Root	Anti venom	Steam distilled	Swiss albino mice and Wistar rats	[20]
*H. indicus*	Root	Acetylcholinesterase Inhibitory Activities	Methanol	Neurotoxic	[21]
*V. planifolia*	Pod	Acetylcholinesterase Inhibitory Activities	Methanol	Neurotoxic	[21]
*D. hamiltonii*	Rhizome	Antimycotic, antiaflatoxigenic and antibiodetriorative activity	Petroleum ether	*A. niger*, *A. columnaris* and *A. tamari*, *F. oxysporum*, *F. proliferatum*, *Drechslera tetramera* and *A. ochraceus*	[22]
*Periploca sepium*	Root bark	Antimicrobial and antioxidant	Hexane, h *n*-hexane-ethyl acetate, acetone	*B. subtilis*, *S. aureus*, *A. tumefaciens*, *E. coli*, *P. lachrymans*, *S. typhimurium*, *X. vesicatoria*	[6]
*Decalepis hamiltonii*	Rhizomes	Antifungal	Petroleum ether	*Alternaria alternata*, *Drechslera tetramera*, *Fusarium oxysporum*, *F. proliferatum*, *Pyricularia oryzae* and *Trichoconis padwickii*	[23]

To overcome these limitations, in vitro culture methods such as callus, cell suspension, and hairy root cultures offer controlled, repeatable platforms for 2H4MB accumulation. Elicitation, precursor feeding, and metabolic manipulation further improve these platforms. At the same time, the potential to create scalable, sustainable, and high-yield production systems is provided by multi-omics analyses, in silico gene cluster prediction, and synthetic biology tools (such as pathway reconstruction, heterologous expression, and CRISPR-mediated metabolic engineering) (Figure 1). The present review discusses about the current developments in synthetic biology tools, metabolic and pathway engineering, plant cell and tissue culture methods, and AI-driven modelling and CRISPR-based genome editing techniques that provide previously unheard-of precision in 2H4MB yield optimisation. This study describes current constraints, unresolved metabolic bottlenecks, and technological gaps by fusing knowledge from conventional in vitro platforms with next-generation computational and genome-engineering tools. Lastly, we suggest future areas of inquiry to build reliable, scalable, and profitable production processes for this valuable aromatic chemical.

## 2. Plant-Level Physiology and Development of 2H4MB

The generation and accumulation of 2H4MB in entire plants are tightly linked to the function and growth of roots, even though the majority of research on this topic has concentrated on enzymes and lab-grown systems. 2H4MB mostly accumulates in the underground part of 2H4MB producing plant such as *Hemidesmus indicus* and *Decalepis species* [7,11]. Rather than being dispersed uniformly throughout the plant, this shows a tight relationship to particular root structures. This trend is comparable to other phenylpropanoid-based secondary metabolites, which frequently accumulate as roots develop, storage tissues form, and carbon is utilised for defence. By altering vascular differentiation, metabolic sink strength, and phenylalanine availability, root growth is known to affect the biosynthesis of secondary metabolites. The idea that 2H4MB accumulation rises with root age and biomass rather than just cellular proliferation is supported by the fact that mature or thicker roots usually show increased phenylpropanoid flow [24,25]. Additionally, elicitation studies that raise 2H4MB levels frequently align with physiological reactions linked to stress, indicating that its synthesis might be synchronised with defensive signals and growth–metabolism trade-offs at the whole-plant level. Despite these findings, there are few direct physiological studies that link 2H4MB concentration to anatomical location, root developmental stage, or overall plant carbon distribution. To ascertain whether 2H4MB is created locally inside specialised root cells or redistributed following its synthesis, future studies combining developmental biology, tissue-specific metabolite profiling, and isotope-based flux analysis will be essential. Applying pathway-level knowledge to sustainable production systems and enhancing cultivation techniques will require these whole-plant viewpoints.

## 3. In Vitro Production of 2H4MB

In vitro culture is crucial for clonal multiplication, plant conservation, and the production of valuable secondary metabolites like 2-Hydroxy-4-Methoxybenzaldehyde (2H4MB). Numerous in vitro methods, such as callus cultures, organogenesis, hairy root cultures, and elicitor-based strategies, have been employed to enhance 2H4MB accumulation under regulated settings. Compared to conventional farming, in vitro systems allow for exact control over critical elements such phytohormones, nutritional composition, temperature, light regime, and elicitor exposure that can significantly impact metabolite production [26]. Furthermore, sterile culture conditions reduce microbial and insect contamination, allowing for the year-round synthesis of metabolites regardless of seasonal or environmental limitations and the manufacture of high-purity chemicals.

Although in vitro cultures provide a powerful platform for enhancing 2H4MB synthesis at the laboratory scale, scaling up from tiny culture volumes to bioreactor-level production remains a special and challenging task. Successful scale-up requires careful optimisation of culture type, oxygen transfer, shear sensitivity, food supply, and elicitor kinetics—all of which may differ dramatically from flask-based systems. Because differentiated cultures, such as adventitious or hairy roots, often exhibit growth and metabolic responses that are extremely system-specific, protocols cannot be easily transferred to large-scale bioreactors [27]. Therefore, even if bioreactor-based culturing shows promise for industrial-level 2H4MB production, further process optimisation and technical research are needed to ensure reproducibility, stability, and economic viability at larger scales. Table 2 shows the 2H4MB production via different in vitro methods.

### 3.1. 2H4MB Production Through Elicitation and Cell Suspension Culture

In plant cell and tissue cultures, elicitation is a potent technique for boosting the biosynthesis of secondary metabolites [36]. The plant’s defence mechanisms are triggered when cultured cells are exposed to particular biotic or abiotic elicitors, which increases the production of target compounds like 2H4MB. By simulating natural stressors, this method activates metabolic processes that would not normally be active under typical growth circumstances. The type, focus, and duration of the elicitor’s exposure, as well as the cultural stage of development, all affect how effective elicitation is. Elicitation has consequently become a dependable and long-lasting technique for raising 2H4MB yields in carefully monitored in vitro settings. Earlier study found that the shikimate and phenylpropanoid are the two major pathways responsible for the biosynthesis of 2H4MB as benzoate pathways originate either directly from shikimate or via phenylpropanoid pathway [33]. Rise in the activities of shikimate dehydrogenase along with phenylalanine ammonia-lyase (PAL) indicate increased accumulation of 2H4MB [12]. A pioneer study aimed to clarify the biosynthesis of 2H4MB in *H. indicus* roots and ascertain how the shikimate pathway contributes to its elicitation-induced production [33]. Prior observations indicated that the phenylpropanoid pathway was the source of the enzymatic pathway leading to the formation of 2H4MB, even though this pathway was previously unknown. To investigate this, the capacity of different elicitors (such as yeast extract, methyl jasmonate, and chitosan) to encourage 2H4MB accumulation was investigated. Yeast extract was the most effective component, which prompted a further in-depth study of metabolic processes. The study looked at the activities of two key enzymes, shikimate dehydrogenase (SKDH) and phenylalanine ammonia-lyase (PAL), as well as the effects of glyphosate inhibition, to see if the shikimate pathway is relevant for 2H4MB production during elicitation [33]. In order to promote metabolic engineering and biotechnological production approaches, this work aimed to better understand the enzymatic and metabolic processes controlling the generation of aroma compounds in *H. indicus* roots.

An efficient in vitro propagation technique for *H. indicus* examined the effects of methyl jasmonate (MeJA) and salicylic acid (SA) on growth and metabolite synthesis [29]. Both elicitors enhanced biomass, antioxidant activity, and metabolite accumulation; however, 75 μM MeJA was the most effective treatment, yielding 3.41 ± 0.8 mg/g DW of 2H4MB, which was slightly higher than the mother plant (3.30 ± 0.2 mg/g DW) and all SA-treated cultures [29]. In *Decalepis salicifolia*, callus cultures were established from root explants, where 84.8% callus induction frequency was achieved on MS medium supplemented with 1.0 μM TDZ and 1.0 μM NAA, indicating the proportion of explants successfully forming callus [10]. Further it was found that chitosan elicitation (200 μM for 72 h) significantly boosted biomass accumulation (9.7 g/L) and raised 2H4MB content to 14.8 μg/g DW, a 1.4-fold increase over the untreated control. Even though the absolute yield was lower than that reported for *H. indicus*, chitosan treatment considerably improved antioxidant capacity (63.8% DPPH RSA and 55.2% HRSA), suggesting its role as a metabolic elicitor [10]. Similarly, *D. hamiltonii* cell suspension cultures were reported to produce 0.92 ± 0.02 mg mL^−1^ (0.092% *w*/*v*) of 2H4MB in the steam-condensed extract obtained from cultured biomass [37].

Precursor feeding is a biotechnological technique that boosts the production of specific secondary metabolites in cell cultures by directly adding precursor molecules to the culture media [38]. For a variety of reasons, precursor feeding in cell suspension cultures is usually restricted to promote the formation of secondary metabolites such as 2H4MB. Optimising the concentrations and timing of precursor addition requires a deep comprehension of the metabolic processes involved. However, little is known about these pathways for the 2HRMB producing plant. Finding suitable precursors that can effectively activate the biosynthetic pathways without causing cytotoxic effects or metabolic abnormalities can also be challenging. Using ferulic acid (FA), the study created a precursor feeding strategy to increase the production of flavour metabolites in *D. hamiltonii* callus suspension cultures [39]. By the fourth week, biomass had reached a peak of 200.38 ± 1.56 g/L, and 2H4MB production had reached 0.08 ± 0.01 mg/100 g DW. Vanillin (0.10 ± 0.02 mg/100 g), 2H4MB (0.44 ± 0.01 mg/100 g), vanillic acid (0.52 ± 0.04 mg/100 g), and ferulic acid (0.18 ± 0.02 mg/100 g) were the results of a significant increase in metabolite accumulation following supplementation with 1 mM FA. The results show that FA feeding efficiently increases flavour metabolites derived from phenylpropanoid, which may lead to large-scale biotechnological production. In a recent study cell suspension culture was established using callus derived from the root and investigated FA and chitosan feeding in *D. hamiltonii* [40]. In comparison to the control, supplementation with 0.1–1.5 mM FA and CH increased biomass and vanillin content by 1.34 and 1.14 times, respectively.

A study aimed to elucidate the biosynthetic origin of 2H4MB in *D. hamiltonii*, based on the notion that it comes via the phenylpropanoid pathway (PPP) [12]. Chemical inhibition tests were conducted to examine route involvement using PPP inhibitors, including propanil, MDCA, and piperonylic acid. Among these, propanil demonstrated a stronger inhibitory impact than MDCA, and both drugs significantly reduced the buildup of 2H4MB (up to 57.9%), indicating the participation of certain PPP branches. Conversely, ferulic acid feeding dramatically boosted the buildup of 2H4MB (up to 107%), encouraging its conversion through downstream PPP intermediates that most likely entailed steps mediated by 4-coumarate: CoA ligase (4CL). Although these conclusions are based on metabolite-level responses rather than transcript-level data, the results provide functional support for PPP involvement in 2H4MB production. These results, when combined with elicitation and precursor-feeding techniques, provide a useful framework for increasing the production of taste metabolites and supporting the conservation-focused use of this threatened species.

The need for additional optimisation is highlighted by the variation in yield between species and elicitor types, even though there has been significant progress in improving 2H4MB production through elicitation and cell suspension cultures. Compared to natural sources, these in vitro methods provide controlled, sustainable, and scalable production; however, there are still drawbacks, including species-specific reactions, low metabolite conversion efficiency, and a lack of knowledge about regulatory enzymes. These gaps might be filled by future research combining metabolic engineering, omics tools, and bioreactor-based scale-up to create a more effective, repeatable system for large-scale 2H4MB biosynthesis.

### 3.2. 2H4MB Production Through Root and Hairy Root Culture

In vitro root and hairy root culture systems can effectively produce secondary metabolites like 2H4MB, providing a sustainable substitute for harvesting endangered species like *D. arayalpathra*, *D. salicifolia*, and *H. indicus*. While in vitro techniques allow for large-scale biomass creation and metabolite production without depleting natural populations, conventional replication of these species by seeds or vegetative means is sometimes constrained by low plantlet yield and susceptibility to disease. *Agrobacterium* rhizogenes-induced hairy root culture provide distinct advantages over adventitious root cultures, including rapid growth, genetic stability, and high metabolite yields that closely resemble the parent plant [41]. This technique creates fast-growing, genetically stable hairy roots by introducing T-DNA from the root-inducing plasmid (Ri plasmid) into plant cells. These cultures offer a reliable, sustainable, and manageable in vitro system by producing large amounts of secondary metabolites under carefully regulated conditions, obviating the need for whole-plant cultivation. Hairy roots are relatively easy to maintain, and elicitors or genetic interventions can be used to precisely control their growth and metabolic pathways [42]. However, the method has drawbacks, such as the initial difficulty of creating cultures varying transformation efficiency based on tissue type and plant species, and sporadic species-specific reactions. Despite these difficulties, hairy root cultures continue to be an effective method for researching plant metabolism and producing high-value secondary metabolites on a large scale. Using juvenile hypocotyls, a study established the *A. rhizogenes*-mediated hairy root cultures in *D. arayalpathra*, with strain TR105 causing the highest frequency (53.2 ± 0.3%) in comparison to cotyledons (32.1 ± 0.2%) [32]. On half-strength MS medium, hairy roots grew quickly from galls, whereas strain MTCC 532 only caused callus. In comparison to normal root cultures (0.16% DW), maximum biomass and 2H4MB accumulation (0.22% DW) were measured during the sixth week. Similarly, the accumulation of 2H4MB and the growth of callus and root tissue were enhanced by optimal hormone combinations of 2,4-D, kinetin, BAP, and NAA as well as combinatorial treatments with thidiazuron (TDZ) and silver nitrate (AgNO_3_) [43]. Within 60 days, adventitious roots in *D. salicifolia* produced 4.9 times more 2H4MB and 35 times more biomass than field-grown plants, according to a recent study [28]. Over the course of two years, steady metabolite accumulation was made possible by ideal circumstances, such as pH and sucrose content. This approach preserves endangered plant populations while offering a scalable and dependable platform for the industrial production of 2H4MB.

Although hairy root cultures are beneficial, maintaining consistent growth, preventing microbial contamination, and guaranteeing consistent metabolite yields are obstacles in large-scale production. By offering a regulated environment for nutrition supply, oxygenation, pH, and other growth parameters, bioreactor systems can assist to get around some of these restrictions and enable scalable and repeatable production. Although their application especially for the synthesis of 2H4MB has not yet been thoroughly investigated, a variety of bioreactor designs such as liquid-phase, gas-phase, or hybrid systems may be modified for hairy root growing [44]. Notwithstanding these benefits, precise optimisation is required because metabolite yields are dependent on several factors, including inoculum density, medium composition, elicitor type, and culture conditions. By combining hairy root culture with elicitation techniques, precursor feeding, and controlled bioreactor systems, 2H4MB can be produced in a sustainable and potentially scalable manner. This is particularly important for meeting industrial demand while preserving endangered plant species. Future research should focus on creating optimal bioprocess conditions and examining metabolic regulation in order to maximise 2H4MB accumulation in vitro.

## 4. 2H4MB Production Through Metabolic Engineering and Genome Mining

A developing multidisciplinary field called “synthetic biology” uses engineering concepts to create and modify biological systems for particular uses. It entails building or altering metabolic pathways and genetic circuits to improve or add desirable functionalities in living organism [45]. Recent developments in this area have revolutionised the production of secondary metabolites, making it possible to develop scalable and effective bioproduction systems for valuable natural chemicals like 2H4MB. In contrast to conventional extraction methods, which rely on slow plant growth and laborious purification, synthetic biology uses genetic engineering, pathway optimisation, and metabolic redesign to build microbial or plant-based “cell factories” that can produce 2H4MB more rapidly and sustainably. This strategy reduces costs and reliance on endangered plants while facilitating the development of novel or rare analogues with enhanced functionality. Additionally, advances in gene cluster analysis and multi-omics have allowed for a better understanding of plant biosynthesis pathways [46]. This has enabled the complete reconstruction of 2H4MB biosynthesis pathways and the logical enhancement of yield and stability through pathway engineering.

Genome mining has emerged as a powerful approach for identifying and exploiting biosynthetic gene clusters responsible for natural compounds such as 2H4MB. By exposing both active and hidden pathways, genome mining provides the genetic template needed to build efficient production systems [47]. Genome mining enables the methodical identification and functional reconstruction of obscure or silent pathways by combining metabolic engineering, bioinformatics, and genome sequencing. This integrated framework offers a revolutionary substitute for conventional extraction-based techniques by improving the yield, diversity, and sustainability of metabolite production in addition to facilitating the logical optimisation of biosynthetic pathways. Important genes and enzymes involved in the biosynthesis of substances like 2H4MB can be found using genome mining, which facilitates the logical design of engineered systems that mimic or improve natural production routes. Additionally, advances in synthetic biology have given scientists unprecedented control over biological systems, enabling the creation of tailored cell factories. Ultimately, a scalable, affordable, and sustainable alternative to the traditional extraction-based synthesis of metabolites obtained from plants is provided by integrating genome mining with modern synthetic biology methods.

The efficient synthetic biology synthesis of secondary metabolites such as 2H4MB requires a number of critical elements. The basis is the identification and characterisation of biosynthetic gene clusters using genome mining, since these clusters contain the genes involved in the production of metabolites [47]. In addition to identifying established pathways, genome mining also identifies quiet or mysterious gene clusters that could be involved in the synthesis of novel metabolites, offering possibilities for further engineering [48]. Exact genetic interventions using CRISPR-Cas9 or other state-of-the-art genome editing technologies enable pathway optimisation and metabolite flux augmentation. Additionally, for efficient metabolite synthesis, the choice of host systems, plant cell cultures, microbial system, or heterologous expression platforms like yeast, *E. coli*, *Nicotiana benthamiana*, or *Arabidopsis thaliana*, must be based on characteristics like metabolic capacity, ease of transformation, and compatibility with target pathways [49,50]. Additional improvements in production efficiency can be achieved through route balancing, bioprocess optimisation (including controlled fermentation or bioreactor operation), and regulatory element fine-tuning. Combining computer modelling and biosensors enables real-time monitoring and adaptive control, ensuring maximum yield and cost-effective efficiency. In order to improve 2H4MB production, genome mining, synthetic biology, and engineered expression systems work together to overcome the limitations of traditional plant extraction methods and enable the discovery of novel or unusual derivatives. This approach is rational, scalable, and sustainable.

### 4.1. 2H4MB Production Through Microbial System

Maintaining enzymatic activity over extended reaction times is crucial for the effective production of 2H4MB. Enzymes frequently show instability and decreased catalytic performance when they are not in their natural settings. This restriction can be effectively overcome by using microbial hosts as heterologous expression platforms, since the cellular environment gives a natural framework for enzyme folding, regeneration, and protection [51]. However, there are still several challenges in implementing microbial production systems for 2H4MB. A significant challenge is the discovery and functional characterisation of key biosynthetic enzymes from source plants, such as *H. indicus*, and the accurate reconstruction of these processes in heterologous microbial hosts. Metabolic flow must also be balanced to ensure an adequate supply of precursors while avoiding an excessive metabolic burden that can hinder host growth and production. Effective cofactor regeneration and redox homeostasis are also significant restrictions because aldehyde production is often intimately associated with NAD(P)H-dependent processes. Furthermore, because aromatic aldehydes like 2H4MB can impede microbial growth and lower fermentation efficiency at high intracellular or extracellular concentrations, product toxicity is a major problem.

Strong precedents for microbial 2H4MB production have been established by the successful production of many aromatic aldehydes that are structurally and biosynthetically related to 2H4MB in microbial systems. For instance, whole-cell biotransformation of trans-anethole using a heterologously expressed trans-anethole oxygenase produced p-anisaldehyde in engineered *Escherichia coli* with a high conversion efficiency of 10.18 mM (1.38 g/L), indicating efficient enzyme stabilisation and cofactor-independent catalysis [52]. An *E. coli* strain known as RARE (Reduced Aromatic Aldehyde Reduction) was developed as a supplemental method to address aldehyde toxicity and reduce it to alcohols. Vanillin and benzaldehyde can build up steadily in this strain. Vanillin synthesis from glucose increased by 55 times as a result [53]. More recently, researchers deleted the aldehyde reductase NCgl0324 from *Corynebacterium glutamicum* [54]. By adding carboxylic acid reductase and O-methyltransferase modules to the route, this modification enhanced the accumulation of 4-hydroxybenzaldehyde (1.36 g/L), protocatechuic aldehyde (1.18 g/L), and vanillin (0.31 g/L). Concurrently, substantial efforts in metabolic engineering have made it possible for bacteria, yeast, and microalgae to produce vanillin [55,56]. Redox state balance, precursor addition, and pathway reconstruction were used to accomplish this. Scalable and sustainable production is supported by these techniques.

The successful synthesis of closely related aromatic aldehydes shows that the main technical obstacles such as enzyme instability, aldehyde reduction, redox imbalance, and pathway flux control can be successfully overcome, despite the lack of a specific microbial biosynthesis system for 2H4MB. Through focused enzyme identification, aldehyde-protective chassis design, and modular pathway engineering, these precedents offer a clear road map for recreating 2H4MB production in microbial hosts. It is anticipated that further integration of these tactics will make microbial production of 2H4MB scalable, controlled, and sustainable. Figure 2 illustrates integrated genome mining, genome editing, and epigenome editing strategies to enhance 2H4MB biosynthesis and accumulation.

### 4.2. Next-Generation Microbial Platforms for 2H4MB Biosynthesis

Biochemical and inhibitor-based studies clearly show that 2H4MB production in *Decalepis* and *Hemidesmus* species originates from the phenylpropanoid pathway, which is provided by upstream phenylalanine generated from shikimate. While increased 2H4MB levels are favourably connected with elicitor-induced PAL activation and glyphosate-sensitive shikimate flow, 2H4MB accumulation is considerably decreased when aminooxyacetic acid inhibits phenylalanine ammonia-lyase (PAL) [44]. Similarly, glyphosate-mediated inhibition of the shikimate pathway decreased 2H4MB accumulation by approximately 40% along with decreased PAL and C2 side-chain cleavage activities, whereas elicitor treatments (yeast extract, chitosan, and methyl jasmonate) markedly increased PAL activity and 2H4MB levels [33]. These quantitative biochemical perturbation studies offer experimentally verified entry points that are directly amenable to metabolic reconstruction in heterologous systems, including PAL, precursors derived from shikimate, and downstream aldehyde-forming steps [53,54,55].

Crucially, it has been repeatedly shown that complex plant secondary metabolic pathways can be successfully transferred into essentially different microbial hosts, setting clear precedents for 2H4MB. Phenylpropanoid-derived aromatic aldehydes such as vanillin have been produced in *Escherichia coli*, *Saccharomyces cerevisiae*, *Corynebacterium glutamicum*, and microalgae through stepwise pathway assembly, cofactor balancing, and suppression of endogenous aldehyde reduction, resulting in gram-per-litre titres (e.g., 0.31–1.36 g/L for vanillin- and hydroxybenzaldehyde-related compounds). Notably, in *C. glutamicum*, deletion of the native aromatic aldehyde reductase NCgl0324 allowed for the stable accumulation of 4-hydroxybenzaldehyde (1.36 g L^−1^), protocatechuic aldehyde (1.18 g L^−1^), and vanillin (0.31 g L^−1^) [54]. This directly demonstrated that host redox control can overcome aldehyde instability, a crucial to potential production of 2H4MB.

Beyond the phenylpropanoid aldehydes, even intricate plant alkaloid biosynthetic pathways have been successfully introduced into microbial systems. As a case in point, the complete biosynthetic routes for the tropane alkaloids hyoscyamine and scopolamine were reconstructed within *Saccharomyces cerevisiae*. This was achieved through the modular expression of over ten plant-derived enzymes, notwithstanding the inherent difficulties associated with compartmentalization and post-translational modifications [56]. Likewise, complex terpenoid and alkaloid pathways, including those for artemisinin precursors and benzylisoquinoline alkaloids (e.g., noscapine), have been reconstituted in yeast [57,58]. In contrast to these systems, the biosynthesis of 2H4MB appears biochemically less complex, primarily depending on cytosolic phenylpropanoid enzymes and aldehyde-forming reactions. These instances, taken together, support the viability of next-generation microbial platforms for 2H4MB, while simultaneously underscoring persisting challenges such as enzyme identification, aldehyde toxicity, and flux optimisation.

Due to the lack of genetic information in *H. indicus* and *Decalepis* plants, the precise biosynthesis mechanism of 2H4MB is still unknown. However, recent developments in genome mining and high-resolution transcriptomics offer a revolutionary route towards this goal [59,60]. Genome mining allows for the systematic identification of candidate biosynthetic gene clusters. This includes those that code for important enzymes, which are expected to be involved in pathways derived from phenylpropanoid or shikimate metabolism [47,61]. By combining machine-learning-based BGC prediction tools (such antiSMASH, PRISM, and DeepBGC) with long-read genome assemblies, researchers can find previously unidentified enzymes that might be involved in the formation of 2H4MB [62,63,64]. Together with structural genes encoding catalytic enzymes, these clusters may also contain transcription factors and regulatory elements that control the dynamics of expression of the 2H4MB biosynthesis pathway. Understanding these cluster organisations can help identify rate-limiting stages and provide insights into the natural regulation of secondary metabolism. Rebuilding or transferring such gene clusters into microbial or plant-based expression systems (e.g., *E. coli*, *Saccharomyces cerevisiae*, or *Nicotiana benthamiana*) can replicate the entire biosynthetic pathway of 2H4MB for sustainable production. This cluster-based approach guarantees efficient substrate channelling and minimal intermediate loss by promoting coordinated gene expression. Synthetic promoters and CRISPR-based regulatory circuits can be used to modify enzyme activity and flux balance in order to further boost yield and stability.

Furthermore, reliable mapping of inducible genes linked to 2H4MB accumulation is made possible by modern transcriptome approaches, including RNA-seq under elicitation, time-series transcriptomics, and single-cell expression profiling [65,66]. Differential expression patterns under stress elicitation, PAL inhibition, or glyphosate treatment can shed light on the regulatory rationale and rate-limiting phases of the system. When combined with co-expression network analysis, multi-omics integration (metabolomics–transcriptomics coupling), and CRISPR-based functional validation in microbial hosts, these techniques can expedite the identification of the complete and authentic 2H4MB biosynthetic pathway [67,68,69]. Ultimately, this modern paradigm transforms a plant-specific secondary pathway into a fully resolved, engineerable metabolic system. In addition to increasing 2H4MB production, these integrated genome-guided and synthetic biology approaches reveal the biochemical and evolutionary processes that underlie the biosynthesis of aromatic compounds. In the end, understanding and modifying these gene clusters can create environmentally benign, scalable systems for the industrial production of 2H4MB, lowering reliance on threatened natural sources like *D. hamiltonii*, *D. arayalpathra* and *D. salicifolia*.

## 5. AI-Assisted Pathway Prediction and Genome Editing Strategies for 2H4MB Enhancement

### 5.1. Pathway Prediction and Optimisation of In Vitro Production System

From a theoretical concept to a useful technique, machine learning (ML) has produced empirically verified insights into poorly understood plant secondary metabolite pathways, setting relevant precedents for the biosynthesis of 2H4MB. The biosynthesis of benzylisoquinoline alkaloids (BIA) in opium poppies (*Papaver somniferum*) is a prominent example of this [70]. In this context, hitherto unknown biosynthetic enzymes were predicted using ML-driven co-expression network analysis, which integrated transcriptomics, metabolite accumulation profiles, and enzyme domain features. The successful clarification of late-stage noscapine biosynthesis processes, including pathway-specific oxidoreductases and acetyltransferases that had resisted traditional biochemistry-based techniques, resulted from the experimental validation of these predictions. Importantly, rather than acting as a post hoc interpretive tool, the ML predictions directly influenced gene selection for functional validation, demonstrating true discovery capabilities as opposed to merely conceptual potential [71].

In another study wherein Artificial neural networks (ANNs) were employed to predict novel candidate genes involved in indole alkaloid biosynthesis, leveraging gene expression profiles and training on known biosynthesis genes [72]. This research illustrates the efficacy of machine learning models in classifying and prioritising genes linked to specialised metabolism, specifically within the intricate pathways of strictosidine and subsequent monoterpene indole alkaloids, based on transcriptome data. Consequently, this approach significantly narrows the scope of candidate genes for subsequent experimental validation. In a recent study on *Spirodela polyrhiza*, a quickly growing aquatic plant regarded as a feasible, sustainable source for natural colours and pharmaceuticals, provides a relevant and experimentally supported example of machine-learning-driven metabolite optimisation [73]. Within this framework, ANNs were effectively employed to model both plant growth and the accumulation of secondary metabolites, whereas a genetic algorithm (GA) was utilised to refine nutrient composition. Relative growth rate and the accumulation of anthocyanins, phenolics, chlorophylls, and antioxidants were accurately predicted by ANN models that included nitrate, phosphate, and ammonium contents. These models showed strong predictive ability (R^2^ > 0.75; MSE < 0.05). With a prediction error of less than 7%, GA-guided optimisation then identified nutritional circumstances that maximised both biomass and metabolite production. This study demonstrates how machine learning may be applied to quantitatively optimise culture conditions for secondary metabolite synthesis in addition to pathway prediction. Similarly, similar approaches based on genetic algorithms and artificial neural networks could be directly applied in 2H4MB-producing plants or heterologous systems to identify the best precursor supply, elicitor combinations, and culture parameters, accelerating yield enhancement even without full pathway elucidation. ML has great potential to optimise 2H4MB synthesis in modified hosts in addition to elucidating pathways [74]. By analysing factors including carbon source availability, media composition, dissolved oxygen, pH, temperature, and elicitor combinations, algorithms trained on bioprocess datasets may determine optimal culture conditions [75]. By fine-tuning fermentation settings or plant cell culture conditions, iterative machine learning-guided optimisation can increase yield while reducing resource inputs. Similar to this, logical metabolic engineering techniques might be guided by ML-based prediction of metabolic bottlenecks or toxicity thresholds, which would reveal which gene modifications or route rewiring steps would best increase 2H4MB flux.

Even though ML has not been applied to *H. indicus* or 2H4MB producing plants in vitro cultures much yet, it is a very relevant technique for boosting 2H4MB production. By combining elicitor-response datasets, culture-performance measures, and omics-derived features, machine learning may be able to identify the environmental cues that most significantly upregulate key steps in the 2H4MB pathway. This predictive capability is a logical connection to genome editing methods since ML-derived insights can identify which genes or regulatory nodes should be modified using CRISPR/Cas systems to further enhance metabolite yields. A powerful, integrated strategy for developing next-generation biotechnological platforms for the synthesis of 2H4MB is offered by targeted gene editing and ML-assisted culture optimisation. Figure 3 depicts the integration of in vitro culture systems with AI/ML-assisted analysis and CRISPR-based genome editing to optimise 2H4MB production and downstream applications.

### 5.2. Genome Editing Strategies for 2H4MB Enhancement

Genome editing is a powerful technique for accurately, efficiently, and consistently rearranging plant metabolic networks and a necessity for boosting the manufacture of specialised metabolites like 2H4MB. Conventional genetic improvement methods, such as gene overexpression or silencing, enabled early developments in the manipulation of chemicals generated from phenylpropanoid [76]. Nevertheless, these techniques were labour-intensive, often imprecise, and unable to consistently target several genes controlling intricate processes. Through contemporary genome editing technologies, a sophisticated toolkit for altering the metabolic architecture supporting 2H4MB production is now accessible.

Among these technologies, CRISPR-based systems provide the highest degree of control because they enable the precise disruption, activation, or replacement of metabolic genes [77]. In order to increase flux through upstream aromatic amino acid pathways, increase the activity of enzymes that limit the rate of 2H4MB biosynthesis, and eliminate competing branches that divert precursors towards unrelated phenolics, genome editing can be utilised. Tightly controlled hydroxylation, methylation, and aldehyde-forming reactions are essential for the biosynthesis of 2H4MB. Because the 2H4MB pathway is controlled by several enzymatic nodes rather than a single regulatory gene, multiplex editing is especially pertinent [78,79,80]. Researchers can reroute carbon flux in the desired direction by creating guide RNAs that concurrently target enzymes in charge of precursor consumption or metabolic bottlenecks. Coordinated editing can reveal hidden regulatory roles and stabilise metabolite output; similar multiplex tactics have been used in other medicinal plants to dismantle redundant gene families and unravel complex phenylpropanoid networks [80,81]. These ideas are immediately applicable to systems that produce 2H4MB, where metabolite growth in vitro is frequently restricted by enzyme redundancy and pathway crosstalk.

For example, CHS2 was knocked out and flavonoid synthesis was decreased by using CRISPR/Cas9 to produce CHS2 mutant cell lines [82]. The metabolic flux was redirected toward stilbenoid production as a result of this downregulation. Resveratrol and its byproducts, such as piceid, consequently rose dramatically. The study shows that CRISPR can improve the generation of important compounds by rerouting metabolic pathways. Similarly, the functional identification of hitherto unidentified secondary metabolites inside the C16 gene cluster was made easier by the use of CRISPR/Cas9 to disrupt biosynthetic genes in *Fusarium graminearum*, indicating their possible role in pathogenicity [83]. Together, these studies show that CRISPR-based genome editing is a versatile method for either reprogramming or exploring metabolic pathways, whether it be by recognising and characterising metabolites associated with disease in plant–pathogen interactions or by rerouting metabolic flux toward beneficial substances like resveratrol. By precisely modifying competing pathways and strategically directing metabolic flux toward particular bio-based chemicals, CRISPR offers significant potential for engineering microbial and plant cell factories to increase the production of industrially significant intermediates, like 2H4MB.

Overexpressing key biosynthetic genes is a straightforward method of increasing flow towards 2H4MB, especially in carefully regulated in vitro systems where precursor supply and enzyme turnover can be properly controlled [84]. Transcription factors (TFs) that regulate the phenylpropanoid and benzenoid pathways, such as the MYB, NAC and WRKY families, are powerful intervention sites in addition to particular enzymes [85,86]. These TFs can collectively improve multiple phases of the 2H4MB pathway, making them great candidates for genome editing. This toolkit is further expanded by epigenome targeted editing, which permits pathway genes to be activated or repressed without changing their DNA sequence [87]. By targeting certain loci to modify methylation or histone marks, CRISPR-based epigenetic modifiers can stabilise the expression of 2H4MB pathway genes that might otherwise fluctuate as a result of environmental or developmental factors [88].

Overall, genome editing offers a powerful route to rationally reprogram the metabolic architecture underlying 2H4MB biosynthesis. Its main advantage is its ability to precisely modify regulatory nodes, like transporters, transcription factors, or structural genes, allowing for targeted enhancement of precursor flux and pathway efficiency. Combining AI-assisted pathway prediction with CRISPR-based editing makes it even more effective, enabling the rapid validation of putative genes and the development of high-yield cell lines tailored for 2H4MB synthesis. Progress is currently hindered by a number of limitations, including incomplete annotation of the enzymes and regulatory factors involved in 2H4MB biosynthesis, the absence of fully resolved genomes for *Hemidesmus*, *Decalepis*, and related aromatic-medicinal species, and the lack of robust transformation and regeneration protocols. These deficiencies account for the absence of documented direct CRISPR uses for 2H4MB improvement. In the future, it will be crucial to develop superior tissue-culture systems, species-specific editing platforms, and high-quality genetic resources. As these technologies develop, it is expected that genome editing, guided by multi-omics, machine learning, and predictive modelling, will be essential to comprehending the metabolic circuitry of 2H4MB and paving the way for next-generation biomanufacturing methods for its large-scale, sustainable production.

Moreover, regulatory and biosafety procedures must be taken into account when using CRISPR and related genome editing technologies to produce 2H4MB in plants. Genome-edited plants intended for use in the food, pharmaceutical, or industrial sectors are subject to containment laws, risk assessment protocols, and species-specific laws. Currently, most studies focus on regulated in vitro or experimental settings, which provide a safe environment for assessing metabolic changes without exposure to the environment. Therefore, as genome editing techniques for *Hemidesmus*, *Decalepis*, and related species advance, future studies aiming at large-scale or commercial production will need to integrate regulatory compliance, risk assessment procedures, and containment strategies alongside metabolic engineering projects.

## 6. Conclusions

This study focusses on recent developments and new approaches for improving the synthesis of 2H4MB, including metabolic engineering, synthetic biology, in vitro plant culture systems, AI-assisted pathway prediction, and CRISPR-based genome editing. While synthetic biology and microbial engineering give chances to reconstruct and optimise biosynthetic pathways outside of the natural plant setting, in vitro methods offer a regulated and sustainable platform for metabolite production. Predictive modelling of metabolic networks is made easier by AI and machine learning, which direct effective pathway optimisation and minimise trial-and-error in culture settings. Targeted improvement of 2H4MB yield is made possible by the precise alteration of structural genes, transcription factors, and regulatory elements made possible by CRISPR and other genome editing technologies.

Despite these advantages, a number of significant disadvantages remain, such as the absence of trustworthy transformation and regeneration processes, the paucity of genomic and transcriptome information for significant aromatic, medicinal species, and the uncertainty surrounding the exact pathway of 2H4MB biosynthesis. These drawbacks explain why many modern methods, particularly CRISPR-based editing and AI-guided engineering, have not yet been directly applied to 2H4MB synthesis.

Future research should focus on generating high-quality genetic resources, elucidating regulatory networks and biosynthetic enzymes, and integrating multi-omics datasets with predictive AI models to guide logical metabolic engineering. With significant ramifications for the flavour, fragrance, and pharmaceutical industries, the integration of sophisticated in vitro systems, genome mining, synthetic biology, and AI-assisted genome editing offers a viable framework for scalable, sustainable, and high-yield production of 2H4MB.

## Figures and Tables

**Figure 1 ijms-27-00503-f001:**
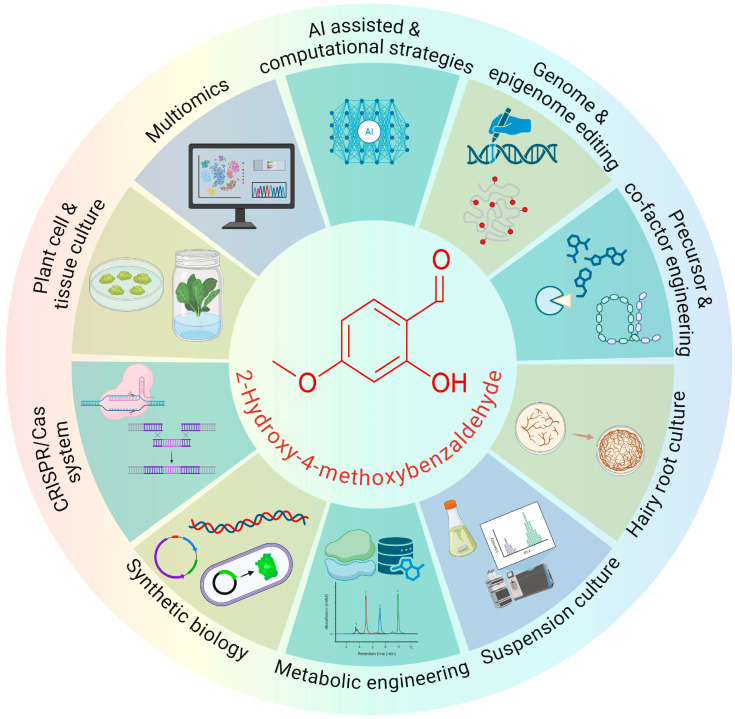
Overview of major biotechnological strategies for enhancing 2H4MB production, including plant cell and hairy root cultures, metabolic and genetic engineering, synthetic biology, and AI-enabled pathway optimisation. Created in BioRender. Ramakrishnan, M. (2026) https://BioRender.com/tomytkh.

**Figure 2 ijms-27-00503-f002:**
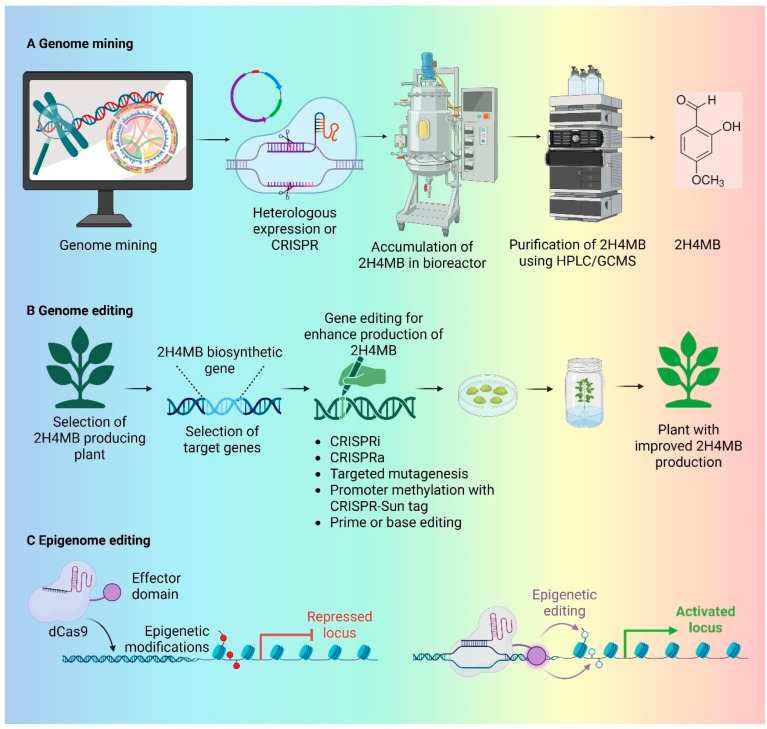
A schematic overview of strategies for improving the production of bioactive compounds, using 2-Hydroxy-4-Methoxybenzaldehyde (2H4MB) as a model. (**A**). Genome mining—Compound accumulation and purification are made possible by the identification of 2H4MB biosynthesis genes, which are then expressed heterologously or through CRISPR-based reconstruction in microorganisms. (**B**). Genome editing: Targeted genome editing in natural plants that produce 2H4MB is demonstrated in this figure. In order to optimise system flow and produce plants with higher 2H4MB production, key biosynthetic or regulatory genes are chosen and altered utilising CRISPR-based techniques (e.g., CRISPRi/a, targeted mutagenesis, base or prime editing, promoter methylation). (**C**). Epigenome editing: This image demonstrates epigenome editing, which modifies epigenetic tags using dCas9 coupled with effector domains. The expression of genes linked to 2H4MB biosynthesis can be accurately regulated by repressing or activating specific loci without changing the DNA sequence. This allows for precise and reversible control over the creation of metabolites. Created in BioRender. Ramakrishnan, M. (2026) https://BioRender.com/av53y62.

**Figure 3 ijms-27-00503-f003:**
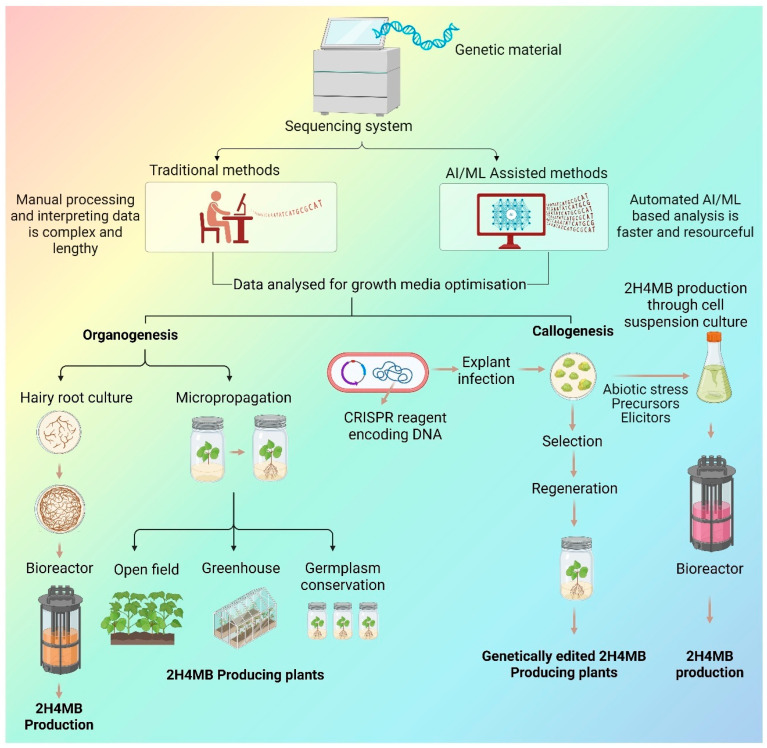
The framework shown in the image is intended to increase the production of 2H4MB by using genetic engineering techniques and in vitro culture methods. Both traditional and AI/ML-assisted sequencing data analysis are used to optimise growth media and culture conditions. Additionally, methods based on organogenesis, like hairy root culture and micropropagation, make it easier to multiply plants that can produce 2H4MB on a big scale. At the same time, callogenesis and CRISPR-based genome editing are used to create genetically modified lines. As a result, these strategies work together to produce 2H4MB in bioreactor systems in a scalable manner. Created in BioRender. Ramakrishnan, M. (2026) https://BioRender.com/fv0wo25.

**Table 2 ijms-27-00503-t002:** 2-Hydroxy-4-Methoxybenzaldehyde (2H4MB) production and its enhancement via in vitro methods.

Plant	Explant	In Vitro/Biological Approaches	Elicitors/Phytohormones/Treatments	2H4MB Content [Reported Yield]	Quantification Method	Reference
*Decalepis salicifolia*	Leaf and Stem	Adventitious root culture	Woody plant medium, NAA (0.5 mg/L), Kn (1.0 mg/L), IBA (0.3 mg/L) and sucrose (2%)	Total production of 2H4MB increases to 4.9-fold (139.54 µg)	RP-HPLC	[28]
*Hemidesmus indicus*	Node, Internode, Leaf	Direct and Indirect Organogenesis	Methyl jasmonate (75µM) and Salicylic acid (2 mg/mL) with BAP (1.5 mg/L), TDZ (2.0 mg/L) and IBA (1.5 mg/L)	Root extracts produced 3.41 mg/g 2H4MB	HPTLC	[29]
*D.* *salicifolia*	Root tuber	Suspension culture (Callus)	Chitosan (CH, 200 µM) and Yeast extract (YE, 200 µM)	Maximum content of 2H4MB at 72 h increases to 1.4-fold (14.8 g/g)	HPLC	[10]
*D. hamiltonii*	Tuberous roots	Developmental (tuber maturation) accumulation	Expression profile of *Dh*PAL, *Dh*C4H, *Dh*COMT and *Dh*VAN)	2H4MB increase with maturation of tuber (10, 170, 500 µg/g in 3-month, 18 month and 60-month-old mature plant, respectively)	HPLC	[30]
*D.* *arayalpathra*	Nodal segment and Shoot tip	Direct organogenesis	Murashige and Skoog’s medium, BA (5.0 μM), IAA (0.5 μM), NAA (2.5 μM) and Adenine sulphate (20.0 μM)	Maximum 2H4MB content which is 9.22 μg/cm^3^ (root extract) recorded	HPLC	[31]
*D. salicifolia*	Nodal segment	Direct organogenesis	Murashige and Skoog’s medium, BA (5.0 μM), IBA (2.5 μM), NAA (0.5 μM) and Adenine sulphate (30.0 μM)	Maximum 2H4MB content which is 6.8 μg/mL (root extract) recorded	HPLC	[24]
*D. arayalpathra*	Cotyledons and Hypocotyls	Hairy root culture	*Agrobacterium rhizogenes* (different strains A4, MTCC 532, TR105 and LBA 5402)	Maximum accumulation of 2H4MB (0.22% dw) recorded at 6th week of growth	TLC	[32]
*H. indicus*	Roots	Roots culture with elicitation	Chitosan, Methyl jasmonate and Yeast extract	Yeast extract for 18 h showed maximum accumulation of 2H4MB (2.7 mg/g)	HPLC	[33]
*H.indicus*	Roots	Roots culture with inhibitor/elicitation treatment	Aqueous Chitosan solution (100 and 200 mg/L) and Aminooxyacetic acid (AOAA) solution (50 and 100 μg/L)	Maximum accumulation of 2H4MB (0.89 mg/g) was detected	TLC	[34]
*H.indicus*	Shoot tip and young roots	Micropropagation (Callogenesis)	Murashige and Skoogs Medium, BA (4.4 µM), IBA (9.8 µM) and NAA (2.69 µM)	Concentration of 2H4MB increases to 2.2-fold (0.12%/g dw)	HPLC	[35]

## Data Availability

No new data were generated in association with this article.

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
