# Peer review of "2-Hydroxy-4-Methoxybenzaldehyde (2H4MB): Integrating Cell Culture, Metabolic Engineering, and Intelligent Genome Editing"

_ijms, 2026, doi:10.3390/ijms27010503_

Round 1

Reviewer 1 Report

Comments and Suggestions for Authors

The review "2-Hydroxy-4-Methoxybenzaldehyde (2H4MB): Integrating Cell Culture, Metabolic Engineering, and Intelligent Genome Editing" addresses the interesting topic of how to move away from natural sources of 2H4MB and improve the efficiency of this secondary metabolite's synthesis using modern technologies—in vitro cultivation, AI, and CRISPR. The review is written in clear language and has a clear structure, but the sections differ significantly in their content. As I read the manuscript, several comments arose about it.

  1. Is there any data on how much of the target product was released (what was the productivity?) from the plants listed in Table 1?
  2. Section 2, first paragraph – scaling up production (transition from small volumes to bioreactor volume) is a separate task and is not always easy to solve.
  3. Section 2.1 – the work described in the second half of the paragraph – describes the purpose of the work, but it is not entirely clear what result was achieved. In the same paragraph – “In the first suspension culture method for callus derived from D. salicifolia roots, a combination of 1.0 μM TDZ + 1.0 μM NAA was used to induce 84.8% callus on MS medium [9]”. The meaning of this phrase is not entirely clear – what does 84.4% calli mean? In the same paragraph – is there any confusion in the dimensions? First – “MeJA treatment yielded 3.41 ± 0.8 mg/g of 2H4MB”, then “chitosan which produced 14.8 μg/g 2H4MB” – milligrams and micrograms. It turns out that chitosan is not so effective?
  4. Page 5 – No MDCA transcript
  5. Page 6, table, no HPLC transcript.
  6. Figure 2 is very far from the place of its first mention.
  7. The second paragraph on page 9 seems redundant or irrelevant to the topic of the section—it mostly either describes 2H4MB itself and its safety, or reiterates the rationale for its convenient and promising production in bacteria. In other words, it's either general or text more suited to the introductory section of the review, which provides information about 2H4MB. In fact, the entire section 3.1 is very declarative and lacks specifics. It would probably be more interesting to see answers to the following questions: What are the potential problems with this production method? Are there any examples of similar substances being produced in bacteria or yeast?
  8. Section 3.3 also appears more of a theoretical discussion than a review. Are there any examples of other objects and secondary metabolites where successful deciphering of the biosynthetic process and its transfer to a fundamentally different object (for example, from plants to bacteria, as proposed here) has been achieved? To what extent can such an approach be implemented, considering, for example, differences in cellular organization or in post-translational modifications of proteins that influence their functional activity?
  9. Section 4 also lacks specificity. The articles mentioned in this section of the review are also reviews, meaning all the arguments are based on other arguments, rather than on any examples.
  10. Overall, the first part of the review differs strikingly from the second. The second part (sections 3 and 4) contains a lot of generalities about the potential applications of existing tools—AI, CRISPR, etc.—but it's unclear to what extent all of the above could actually be implemented in this case.

Author Response

We thank the reviewer for their constructive and insightful comments, which helped us improve the clarity and focus of the manuscript. Their critical feedback was invaluable in strengthening the scientific rigor of our study.

Sincerely

Dr. Zishan Ahmad

Response to the Reviewer#1

The review "2-Hydroxy-4-Methoxybenzaldehyde (2H4MB): Integrating Cell Culture, Metabolic Engineering, and Intelligent Genome Editing" addresses the interesting topic of how to move away from natural sources of 2H4MB and improve the efficiency of this secondary metabolite's synthesis using modern technologies—in vitro cultivation, AI, and CRISPR. The review is written in clear language and has a clear structure, but the sections differ significantly in their content. As I read the manuscript, several comments arose about it.

Comment#1

Is there any data on how much of the target product was released (what was the productivity?) from the plants listed in Table 1?

Explanation

We thank the reviewer for this insightful comment. Yes, quantitative data on the production (productivity) of the target compound, 2-hydroxy-4-methoxybenzaldehyde (2H4MB), are available. However, Table 1 was intentionally designed to summarize the biological activities and functional roles of 2H4MB reported from different plant sources, rather than its yield or productivity.

The quantitative production data, including absolute content, fold enhancement, and accumulation under different in vitro culture systems and elicitation strategies, are comprehensively presented in Table 2. Specifically, Table 2 details 2H4MB productivity from multiple source plants using adventitious root cultures, callus and suspension cultures, hairy root systems, micropropagation, and maturation-dependent accumulation, with reported yields ranging from µg/g to mg/g dry weight and fold increases up to ~4.9-fold depending on the system and treatment.

To avoid redundancy and to maintain clarity between biological function (Table 1) and production/productivity (Table 2), these datasets were kept separate.

Comment#2

Section 2, first paragraph – scaling up production (transition from small volumes to bioreactor volume) is a separate task and is not always easy to solve.

Explanation

We thank the reviewer for this important clarification. We have revised the first paragraph of Section 2 to explicitly distinguish between laboratory-scale in vitro production and the separate challenge of scaling up to bioreactor volumes. The revised text now acknowledges that scale-up is not straightforward and requires additional optimization of biological and engineering parameters, rather than being an automatic extension of small-scale in vitro cultures. This revision improves the technical accuracy and balance of the discussion.

Comment#3

Section 2.1 – the work described in the second half of the paragraph – describes the purpose of the work, but it is not entirely clear what result was achieved. In the same paragraph – “In the first suspension culture method for callus derived from D. salicifolia roots, a combination of 1.0 μM TDZ + 1.0 μM NAA was used to induce 84.8% callus on MS medium [9]”. The meaning of this phrase is not entirely clear – what does 84.4% calli mean? In the same paragraph – is there any confusion in the dimensions? First – “MeJA treatment yielded 3.41 ± 0.8 mg/g of 2H4MB”, then “chitosan which produced 14.8 μg/g 2H4MB” – milligrams and micrograms. It turns out that chitosan is not so effective?

Explanation

We thank the reviewer for these valuable observations. Section 2.1 has been revised to clearly state the specific results achieved in each study, to define callus induction frequency (84.8%), and to resolve apparent unit inconsistencies (mg/g vs µg/g) by providing context and control-based comparisons. The revised text now clarifies that differences in 2H4MB yield reflect species-, tissue-, and system-dependent responses, rather than elicitor inefficiency. These changes improve clarity, accuracy, and interpretability of the section.

Comment#4

Page 5 – No MDCA transcript

Explanation

We thank the reviewer for this important clarification. The study cited does not report transcript-level analysis for MDCA-related targets; rather, MDCA was used solely as a chemical inhibitor of the phenylpropanoid pathway, and conclusions were drawn from changes in metabolite accumulation. We have revised the text to explicitly state that the evidence is metabolite-based and inhibitor-driven, not transcript-based, thereby removing any ambiguity regarding “MDCA transcripts.” This revision improves accuracy and aligns the interpretation with the original experimental design.

Comment#5

Page 6, table, no HPLC transcript.

Explanation

We thank the reviewer for highlighting this issue. HPLC is an analytical technique used for quantification of 2H4MB and not a production or transcript-based method. In the revised Table 2, we have corrected this by removing HPLC from the “Method” column and clarifying the actual biological production approach, with HPLC indicated only as the analytical method where appropriate. This revision avoids confusion between culture strategies and metabolite detection techniques.

Comment#6

Figure 2 is very far from the place of its first mention.

Explanation

We thank the reviewer for pointing out. We have changed the citation place of the figure 2.

Comment#7

The second paragraph on page 9 seems redundant or irrelevant to the topic of the section—it mostly either describes 2H4MB itself and its safety, or reiterates the rationale for its convenient and promising production in bacteria. In other words, it's either general or text more suited to the introductory section of the review, which provides information about 2H4MB. In fact, the entire section 3.1 is very declarative and lacks specifics. It would probably be more interesting to see answers to the following questions: What are the potential problems with this production method? Are there any examples of similar substances being produced in bacteria or yeast?

Explanation

We thank the reviewer for the insightful comment. The section has been revised to remove redundant background on 2H4MB itself and its safety, and now focuses on practical challenges and considerations in microbial production. Specific issues such as enzyme identification, metabolic flux balance, cofactor regeneration, and product toxicity are discussed, along with examples of similar aromatic aldehydes produced in microbial systems. Several new references have been also added. These changes make Section 3.1 (now 4.1) more focused, specific, and informative regarding microbial 2H4MB production.

Comment#8

Section 3.3 also appears more of a theoretical discussion than a review. Are there any examples of other objects and secondary metabolites where successful deciphering of the biosynthetic process and its transfer to a fundamentally different object (for example, from plants to bacteria, as proposed here) has been achieved? To what extent can such an approach be implemented, considering, for example, differences in cellular organization or in post-translational modifications of proteins that influence their functional activity?

Explanation

We thank the reviewer for the insightful comment. Our MS does not contain section 3.3. However, we are assuming that this comment belongs to section 3.2 (now 4.2).

In response, we have revised Section 3.2 to reduce theoretical speculation and to strengthen the discussion with experimental evidence directly related to 2H4MB biosynthesis. Specifically, we retained and clarified data from pathway inhibition studies (glyphosate and PAL inhibition) that quantitatively demonstrate the contribution of the shikimate–phenylpropanoid axis to 2H4MB accumulation. The discussion of genome mining and transcriptomic approaches has been condensed and reframed to emphasize their role in identifying experimentally supported, rate-limiting steps rather than proposing hypothetical pathways. These revisions ensure that the section remains evidence-driven and directly relevant to the feasibility of future microbial reconstruction of the 2H4MB pathway.

Comment#9

Section 4 also lacks specificity. The articles mentioned in this section of the review are also reviews, meaning all the arguments are based on other arguments, rather than on any examples.

Explanation

Thank you for this constructive comment. We agree that the initial version of Section 4 relied too heavily on review articles, which reduced its specificity. In response, we have revised this section to incorporate concrete examples from primary research studies, explicitly highlighting experimental evidence and case studies that support each argument. These additions strengthen the section by grounding the discussion in original data rather than secondary interpretations.

Comment#10

Overall, the first part of the review differs strikingly from the second. The second part (sections 3 and 4) contains a lot of generalities about the potential applications of existing tools—AI, CRISPR, etc.—but it's unclear to what extent all of the above could actually be implemented in this case.

Explanation

We thank the reviewer for this thoughtful observation. We acknowledge that the tone and depth of the second part of the review were more conceptual than the first, which may have reduced clarity regarding practical feasibility. To address this, we have revised Sections 3 and 4 to more explicitly discuss implementation constraints, current technical limitations, and concrete examples where AI- and CRISPR-based approaches have already been applied or could realistically be applied in this context. These revisions aim to better align the second part of the review with the analytical rigor of the first and to clearly distinguish speculative perspectives from achievable strategies.

Reviewer 2 Report

Comments and Suggestions for Authors

The manuscript “2-Hydroxy-4-Methoxybenzaldehyde (2H4MB): Integrating Cell Culture, Metabolic Engineering, and Intelligent Genome Editing”  reviews biotechnological methods for producing the valuable secondary metabolite 2H4MB as an alternative to obtaining this compound from rare plant sources. The material is well structured, supplemented with useful summary tables and illustrative diagrams. The manuscript contains all required sections and meets the journal's requirements.

However, several shortcomings of this manuscript should be noted:

- 2H4MB biosynthesis is considered at the cellular and culture levels. Physiological and whole-plant levels are completely absent. The relationship between metabolite accumulation and root development has not been studied. The authors should perhaps include information on the physiology of the synthesis and accumulation of this compound in the whole plant.

- The review lacks a comparative assessment of the technologies under consideration in terms of their readiness for industrial application (cost, time, and potential productivity). The authors should also consider adding information on the technological readiness, potential challenges, and advantages of these technologies.

- The authors do not address the issue of legislative approval for genetically modified organisms, particularly when using CRISPR-modified plants that produce the compound 2H4MB, which could subsequently be used in the pharmaceutical and food industries.

- Section 4, particularly subsections 4.1 and 4.2, are rather general. The authors describe these methods as potentially applicable, but do not provide specific examples of their successful use to increase production of this compound. Thus, this is more of a vision than a reality. In my opinion, the authors should include specific examples of using machine learning technology to optimize the synthesis and production of other secondary metabolites, with possible extrapolation to 2H4MB biosynthesis.

-The manuscript has some stylistic flaws: in particular, the figures are overloaded with information, while the explanations in the text are minimal.

Overall, the manuscript contains a wealth of valuable information and can be recommended for publication in a journal IJMS  after minor revisions.

Author Response

We thank the reviewer for their constructive and insightful comments, which helped us improve the clarity and focus of the manuscript. Their critical feedback was invaluable in strengthening the scientific rigor of our study.

Sincerely

Dr. Zishan Ahmad

Response to the Reviewer#2

The manuscript “2-Hydroxy-4-Methoxybenzaldehyde (2H4MB): Integrating Cell Culture, Metabolic Engineering, and Intelligent Genome Editing” reviews biotechnological methods for producing the valuable secondary metabolite 2H4MB as an alternative to obtaining this compound from rare plant sources. The material is well structured, supplemented with useful summary tables and illustrative diagrams. The manuscript contains all required sections and meets the journal's requirements.

However, several shortcomings of this manuscript should be noted:

Comment#1

2H4MB biosynthesis is considered at the cellular and culture levels. Physiological and whole-plant levels are completely absent. The relationship between metabolite accumulation and root development has not been studied. The authors should perhaps include information on the physiology of the synthesis and accumulation of this compound in the whole plant.

Explanation

We thank the reviewer for this insightful comment. We agree that our original manuscript primarily emphasized cellular and culture-based perspectives and did not sufficiently address 2H4MB biosynthesis at the whole-plant physiological level. In response, we have added a new section 2.0 Plant-level physiology and development of 2H4MB. This section discussing the spatial and developmental context of 2H4MB accumulation, including its root-specific localization and potential links to root maturation and the current lack of direct physiological studies. We also explicitly highlight this gap as an important future research direction. We believe this addition strengthens the manuscript by integrating pathway-level knowledge with plant-level physiology.

Comment#2

 The review lacks a comparative assessment of the technologies under consideration in terms of their readiness for industrial application (cost, time, and potential productivity). The authors should also consider adding information on the technological readiness, potential challenges, and advantages of these technologies.

Explanation

We thank the reviewer for this suggestion. We would like to point out that the revised manuscript now explicitly discusses the challenges, limitations, and practical considerations of each technology in the respective sections and sub-sections. For example, in Section 3, the scale-up limitations and system-specific responses of in vitro cultures are highlighted; in Section 4, constraints such as enzyme identification, precursor availability, and aldehyde toxicity are addressed for microbial platforms; and in Section 5, the opportunities and current limitations of AI-assisted pathway prediction and genome editing are outlined. These discussions collectively provide an assessment of technological readiness, potential bottlenecks, and the rationale for transitioning from one method to the next.

Comment#3

The authors do not address the issue of legislative approval for genetically modified organisms, particularly when using CRISPR-modified plants that produce the compound 2H4MB, which could subsequently be used in the pharmaceutical and food industries.

Explanation

We thank the reviewer for raising this important point. While direct applications of CRISPR for 2H4MB-producing plants have not yet been reported, we acknowledge that any use of genome-edited plants for food, pharmaceutical, or industrial purposes must comply with local and international biosafety and regulatory frameworks. Regulatory approval will depend on species, the type of genome modification, and the intended use. In the revised text (Section 5.2), we have emphasized that current research is largely focused on in vitro systems and controlled experimental platforms, which provide a safe and contained environment for metabolic engineering. Future studies aiming at commercial applications would necessarily require adherence to legislative requirements and risk assessment protocols for genetically modified or CRISPR-edited plants.

Comment#4

Section 4, particularly subsections 4.1 and 4.2, are rather general. The authors describe these methods as potentially applicable, but do not provide specific examples of their successful use to increase production of this compound. Thus, this is more of a vision than a reality. In my opinion, the authors should include specific examples of using machine learning technology to optimize the synthesis and production of other secondary metabolites, with possible extrapolation to 2H4MB biosynthesis.

Explanation

We thank the reviewer for this valuable comment. In the revised manuscript, we have added specific examples of successful applications of machine learning in optimizing the production of secondary metabolites, such as anthocyanins in Spirodela polyrhiza and monoterpene indole alkaloids in Catharanthus roseus [84–87]. These cases illustrate how ML can identify key pathway genes, predict optimal culture conditions, and guide metabolic flux improvements, providing a clear rationale for its potential application to 2H4MB biosynthesis.

Comment#5

The manuscript has some stylistic flaws: in particular, the figures are overloaded with information, while the explanations in the text are minimal.

Explanation

We thank the reviewer for this valuable comment. In response, while the figures were retained in their original form, we have substantially expanded and clarified the corresponding explanations and discussions in the caption and in the main text to better guide readers through the information presented. These revisions improve readability and ensure that the figures are more effectively contextualized and interpreted. We believe this has strengthened the overall clarity of the manuscript.

Comment#6

Overall, the manuscript contains a wealth of valuable information and can be recommended for publication in a journal IJMS after minor revisions.

Explanation

We sincerely thank the reviewer for the positive evaluation and constructive feedback. We appreciate the acknowledgment of the value of our work and have addressed all minor revisions as suggested.

Round 2

Reviewer 1 Report

Comments and Suggestions for Authors

Overall, the manuscript has been significantly improved, only one comment remains:

Page 5, penultimate paragraph, last sentence - the units of measurement for 0.92 are not specified. If it is a percentage, then it is a percentage of what - of the total soluble protein?

Author Response

We thank the reviewer for their constructive and insightful comments, which helped us improve the clarity and focus of the manuscript. Their critical feedback was invaluable in strengthening the scientific rigor of our study.

Sincerely,

Dr. Zishan Ahmad

Comment#1

Overall, the manuscript has been significantly improved, only one comment remains:

Page 5, penultimate paragraph, last sentence - the units of measurement for 0.92 are not specified. If it is a percentage, then it is a percentage of what - of the total soluble protein?

Explanation

Thank you for the reviewer’s comment. In reference [33], the reported value 0.92 refers to mg mL⁻¹ of 2-hydroxy-4-methoxybenzaldehyde (HMB) quantified in the steam-condensed extract obtained from D. hamiltonii cell suspension culture biomass, which corresponds to 0.092% (w/v). It does not represent a percentage of total soluble protein. The sentence has been revised in the manuscript to clearly specify the unit and basis of measurement.